# Autophagy and Apoptosis: Current Challenges of Treatment and Drug Resistance in Multiple Myeloma

**DOI:** 10.3390/ijms24010644

**Published:** 2022-12-30

**Authors:** Omar S. Al-Odat, Daniel A. Guirguis, Nicole K. Schmalbach, Gabriella Yao, Tulin Budak-Alpdogan, Subash C. Jonnalagadda, Manoj K. Pandey

**Affiliations:** 1Department of Biomedical Sciences, Cooper Medical School of Rowan University, Camden, NJ 08103, USA; 2Department of Chemistry and Biochemistry, Rowan University, Glassboro, NJ 08028, USA; 3Department of Hematology, Cooper Health University, Camden, NJ 08103, USA

**Keywords:** multiple myeloma (MM), autophagy, apoptosis, bone marrow microenvironment (BMM), drug resistance, anti-apoptotic proteins

## Abstract

Over the past two decades, the natural history of multiple myeloma (MM) has evolved dramatically, owing primarily to novel agents targeting MM in the bone marrow microenvironment (BMM) pathways. However, the mechanisms of resistance acquisition remain a mystery and are poorly understood. Autophagy and apoptosis are tightly controlled processes and play a critical role in the cell growth, development, and survival of MM. Genetic instability and abnormalities are two hallmarks of MM. During MM progression, plasma malignant cells become genetically unstable and activate various signaling pathways, resulting in the overexpression of abnormal proteins that disrupt autophagy and apoptosis biological processes. Thus, achieving a better understanding of the autophagy and apoptosis processes and the proteins that crosslinked both pathways, could provide new insights for the MM treatment and improve the development of novel therapeutic strategies to overcome resistance. This review presents a sufficient overview of the roles of autophagy and apoptosis and how they crosslink and control MM progression and drug resistance. Potential combination targeting of both pathways for improving outcomes in MM patients also has been addressed.

## 1. Introduction

Lymphoma, leukemia, and myeloma are the three main classifications for cancers of the hematopoietic system. Multiple myeloma (MM) is caused by aberrant plasma cells (PCs) in the bone marrow (BM), representing 1% of neoplastic diseases and 13% of hematological neoplasms [1]. MM is a challenging cancer to diagnose and treat. Patients go through two early stages of the disease before acquiring aggressive myeloma. Monoclonal gammopathy of undetermined significance (MGUS) and smoldering multiple myeloma (SMM) are the precursor stages with no clinical symptoms defined (Figure 1). Depending on the stage of the disease, the number of abnormal PCs, levels of monoclonal immunoglobulin (Ig), and cytogenetic abnormalities begin to rise, and ultimately patients become symptomatic (hypercalcemia, anemia, renal insufficiency, bone lesions, and multiple infections), and referred as MM. The American Cancer Society estimates that by the year 2022, approximately 34,470 new cases of MM will be diagnosed and 12,640 deaths from MM in the United States will occur [2].

The interaction of myeloma cells to bone marrow microenvironment (BMM) is the hallmark of MM [3]. Although treatment options for MM have improved remarkably over the last several decades, the survival rate remains extremely low, and all patients experience relapse or become refractory. Unsurprisingly, refractory and relapsed (R/R) patients often present with a more aggressive phenotype upon recurrence. It has been established that cytogenetic and epigenetic abnormalities play critical role in MM progression and resistance to available drugs. However, defective mechanisms of autophagy, apoptosis, tumor microenvironment, and cell survival signaling contribute in MM progression and drug resistance. Table 1 illustrates the common drugs that have been used to treat MM patients, its mechanism of action, and underlying mechanisms of drug resistance.

## 2. Dual Role of Autophagy in Cancer

Autophagy is a cellular catabolic process, highly conserved from yeast to mammals [48,49]. Autophagic “self-eating” is a natural recycling and cleaning program, which breakdown unneeded or damaged components within a cell and reuse it for cellular repair or regenerating newer and healthier cells. Interestingly, the processes of autophagy may be either selective or non-selective (general recycling). The general recycling non-selective autophagy route is mediated by the creation of a vesicular structure called an autophagosome. In contrast, selective autophagy has a larger role in immunity because it can detect targets, such as intracellular pathogens, misfolded protein accumulations and damaged cell organelles. Autophagy in eukaryotic cells is categorized into three main types including microautophagy, chaperone-mediated autophagy (CMA), and macroautophagy (Figure 2). Although the three types of autophagy are mechanistically different, they all involve the process of lysosomal degradation [48,50,51].

In microautophagy, the lysosome engulfs the autophagic cargo directly; the lysosomal membrane first invaginates, forming a bud and sequestering the cytoplasmic material. Then, the invaginated membrane pinches off inside the lumen to form a small vesicle [52]. CMA is primarily activated under stressful conditions such as starvation. It is peculiar as it targets cytosolic proteins with a similar sequence to Lys-Phe-Glu-Arg-Gln (KFERQ) consensus motif [53]. First, the protein with the KFERQ consensus motif is recognized by heat shock cognate protein 70kDa (hsc70) and other proteins. This complex also helps unfold and deliver the protein to the lysosomal-associated membrane protein 2A (LAMP2A). LAMP2A then multimerizes, upon binding to the protein, a step required for the translocation of the substrate protein. Another form of the hsc70 found in the lumen of the lysosome called lysosomal hsc70 (lys-hsc70) helps disassemble the LAMP2A from the multimeric complexes once the cargo protein enters the lumen of the lysosome. CMA and macroautophagy often work sequentially. For example, macroautophagy is activated during acute starvation, but as starvation persists, CMA gets activated to produce necessary amino acids for protein synthesis [54].

Macroautophagy is the most researched type of autophagy. It involves de novo cytosolic double/multi-membrane autophagosome formation to sequester and transport various cellular components to the lysosome [51,55]. Macroautophagy is the primary process that delivers various cytoplasmic constituents to the lysosome; therefore, it is often referred to simply as autophagy [56]. Macroautophagy involves multiple sequential steps. The first step is initiation which is linked to the endoplasmic reticulum (ER) at a subdomain called the omegasome, which is enriched by phosphatidylinositol 3-phosphate (PI(3)P). The second step involves forming a cup-shaped phagophore, which continues to elongate to surround the cellular material. As the phagophore closes upon itself and the tip fuses together, it becomes an autophagosome. Finally, the autophagosome sometimes fuses with an endosome and ultimately merges with the lysosome, creating an autolysosome. The sequestered material is degraded and released into the cytoplasm [57,58]. Macroautophagy in mammalian cells is controlled by many signaling pathways, each of which is responsible for a unique aspect of the autophagy process. Induction step is regulated by a complex composed of various proteins including UNC-51-like autophagy-activating kinase (ULK1/2), (autophagy-related protein 13) ATG13, RB1-inducible coiled-coil 1 (RB1CC1, also known as FIP200), and C12orf44/ATG101. This complex is controlled by the mammalian target of rapamycin (mTOR), which functions as a negative regulator of autophagy by phosphorylating autophagy-related protein (ATG). Inhibition of mTORC1, one of the main mTOR complexes by stressful conditions, p53, and AMP-activated protein kinase (AMPK) complex leads to the dephosphorylation and activation of ULK complex, which localizes the complex to phagophore, triggering the nucleation complex phosphorylating class III Phosphatidylinositol-3 kinase (PI3K class III). The nucleation step of the phagophore requires a complex PI3K class III composed of VPS34, VPS15, and Beclin1, ATG14, and Autophagy and Beclin1 Regulator 1 (AMBRA1). AMBRA1 induces macroautophagy, while anti-apoptotic proteins inhibit it by binding to Beclin1. This complex is essential as it produces. PI (3)P is needed in the macroautophagy process. The maturation and elongation step of the phagophore is accomplished with the help of several interactions between autophagy-related proteins ATG12-ATG5-ATG16L1 conjugation complex, and microtubule-associated protein-1 light chain kinase 3 (LC3) conjugation system, which aid in phagophore expansion shadowed by elongation [51,58]. LC3 II allows autophagosomes to attach to degraded substances such as misfolded proteins and damaged organelles, following the formation of autophagosomes. Multiple SNARE-like proteins regulate the fusion of autophagosomes and lysosomes, resulting in autolysosomes that selectively remove proteins and other damaged organelles.

Autophagy is involved in various biological processes, including turnover of organelles and intracellular proteins, production of amino acids during starvation, degradation of bacteria (xenophagy), antigen presentation to major histocompatibility complex class II (MHC class II), and type 2 cell death/autophagic cell death [56,57]. In the absence of stressful conditions, the basal autophagy level is usually low, and it functions as a housekeeper by eliminating damaged proteins and organelles and maintaining homeostasis. On the other hand, under stressful conditions such as starvation, autophagy is induced to generate the necessary amino acids, regain homeostasis, and promote cell survival [49,56,59,60]. Importantly, dysregulation of autophagic genomic damage, and metabolic stress can lead to various diseases including neurodegeneration, aging, autoimmune diseases, and cancer [61,62,63]. However, the precise role is still hotly contested and fraught with ambiguity.

Interestingly, autophagy has dual role in cancer. Disrupted autophagy has been reported in many solid tumors such as colorectal, pancreatic, hepatocellular, and cholangiocarcinoma, as well as hematological malignancies such as leukemia, lymphoma, and myeloma. Moreover, alteration in autophagy can develop drug resistance following several chemotherapy treatments including cisplatin, paclitaxel, gemcitabine, etoposide, and doxorubicin [64]. However, the role of autophagy in cancer is not fully understood since it has both oncogenic and tumor-suppressive effects [57,60,65,66]. Evidence supports the idea that autophagy acts as a tumor suppressor. Tumors often lack genes required for autophagy. For instance, human breast cancer has frequent monoallelic deletions and decreased expression of Beclin1, which is a gene that encodes an essential protein component of the PI3K complex [67]. Another report shows that Beclin1 heterozygous disruption increases the development of premalignant lesions induced by the hepatitis B virus and increases the frequency of spontaneous malignancies [68]. Furthermore, it has been suggested that Beclin1 and its positive regulator coiled-coil UV radiation resistance-associated gene (UVRAG) are important in promoting autophagy and suppressing proliferation and tumorigenesis. UVRAG suppressed the proliferation rate, tumorigenicity, and anchorage-independent growth of HCT116 human colon cancer cells, which contain a monoallelic frameshift mutation that creates a premature stop codon in the UVRAG gene [69].

Autophagy contributes to tumorigenesis as it helps the tumor cell adapt to the stressful environment, such as ischemic conditions, which tend to be localized in the center of tumors and provide the tumor with the necessary nutrients for growth and proliferation until angiogenesis is established [70]. Preclinical studies in mouse models of human breast cancer identified the pro-tumorigenesis function of FIP200. This study showed that the deletion of FIP200 in mammary tumor cells inhibits tumor initiation and progression [71]. Moreover, it has been established that cytoplasmic liver kinase B1 (LKB1) enhances autophagy and induces the growth of lung adenocarcinoma [72]. In addition, autophagic activity is greatly reduced in malignant cells due to the activation of Akt after the deletion of the tumor suppressor phosphatase and tensin analog (PTEN) [73]. Autophagy is associated with the transcription of genes regulated by hypoxia-inducible factor-1 (HIF-1). As indicated by the fact that hypoxia-induced AMPK adversely affects mTOR signaling, it ultimately results in ULK1 phosphorylation and the initiation of autophagy [74]. In conclusion, these results show that autophagy-targeting drugs and gene knockdown therapies may kill cancer cells.

Autophagy acts as a double-edged sword in cancer cells. It could either promote tumor formation, proliferation, and metastasis or serve as a mechanism of tumor suppression. Therefore, treatment strategies targeting autophagy focus on two principles: First inhibiting autophagy to eliminate its tumor-protective effects, which would entail the degradation of misfolded and unfolded proteins and reducing energy supply. The second principle is to induce autophagy to promote autophagic cell death. Under drug treatment, autophagy acts as a pro-survival mechanism. An in-depth familiarity with the fundamental cellular mechanisms that govern autophagy in both normal and tumor cells is necessary for the development of novel agents that specifically target autophagy in cancer cells.

Since autophagy’s role in cancer and many other diseases is prominent, researchers have developed various drugs that either induce or attenuate autophagy’s activities. The most common way of studying a protein’s cellular function requires an indirect method, such as knockout or deletion mutations to inhibit or oncogenic mutations that activate genes that encode the specific proteins of interest. Using small molecules, known as chemical genetics, is a direct and complementary approach that allows selective activation or inhibition of the protein of interest [75]. There are various small molecule activators and inhibitors of autophagy that were developed to treat different human diseases. Autophagy activators include inhibitors of mTOR signaling, activators of AMPK, inhibitors of class I PI3K signaling, Inositol trisphosphate (IP3) inhibitors, the epidermal growth factor receptor (EGFR)/ receptor tyrosine kinase inhibitors, and small molecule mimetics BH3 which block interaction between anti-apoptotic proteins and Beclin1. Autophagy inhibitors include inhibitors of VPS34 kinase complexes, an inhibitor of ubiquitin-specific peptidases that regulate the stability of Beclin1 and VPS34, PI3K inhibitors that specifically target class III PI3K, lysosomal alkalizers, ATPase inhibitors, vacuolar H⁺-ATPases inhibitors, compounds that interfere with the fusion of the autophagosome and lysosome (protein transport inhibitors), inhibitors of lysosomal hydrolases, and proteases. Modulating intracellular Ca²⁺ and cytoskeletal components such as microtubules or microtubule-associated proteins can activate or inhibit autophagy. Table 2 illustrates the mechanism of action of the most prominent autophagy inhibiting and inducing drugs that have been used to treat diseases.

## 3. Autoghagy in MM

Among the many essential biological functions of autophagy, it plays a critical role in innate and adaptive immunity. Autophagy helps eliminate various microorganisms, controls pro-inflammatory signaling and secretion of immune mediators, and controls homeostasis of lymphocyte and antigen presentation [76]. B lymphocytes include two distinct populations known as B1 and B2 cells, which vary distinctively in function and development. The B1 cells are involved in the innate immune system as opposed to the B2 cells, which are considered part of the adaptive immune system [77]. Autophagy is not necessary for B1 and B2 cell development but is required for B1 self-renewal in the periphery. Moreover, autophagy plays a vital role in the B2 cell’s Toll-like receptor (TLR) activation [78]. Pengo et al. demonstrated thatPCs have high autophagic activities and suggested that autophagy has a general role in PC oncogenesis. They also showed the viability of PCs as well as the importance of autophagy in sustaining antibody responses and its necessity in long-lived BM PCs [79]. Furthermore, Halliley et al. were able to demonstrate the importance of autophagy as a mechanism of survival in human long-lived PCs [80].

MM has similarities with the original plasma cell clone, such as an enlarged ER and cytoplasm, as well as immunoglobulin (Ig) secretion. MM cells produce a considerable amount of Ig, inevitably producing misfolded or unfolded proteins that can be toxic to the cells. MM uses various molecular pathways to protect the cells from damage and promote cell survival, including processes such as proteasomal degradation, UPR, and autophagy. In cases where there is excessive production of misfolded protein and the inability of the proteasomal system or UPR to deal with this stress, autophagy could be induced to help alleviate the stress. If all these survival mechanisms fail, apoptosis is engaged, and the cells get destroyed [81,82,83].

Autophagy in myeloma cells not only works in tandem with the ubiquitin-proteasome system (UPS) to remove ubiquitinated proteins [84], but it also plays a role in determining the susceptibility to proteasome inhibitors [85]. According to recent research on MM, greater immunoreactivity against autophagic markers including Beclin1 and LC3 corresponds with longer patient survival [86]. Furthermore, it has been shown that the imbalance between Beclin1 and p62 proteins expression promotes the proliferation of MM cells through autophagy regulation [87].

Aberrant DNA repair mechanisms contribute to illness development, initial translocations, and progression in MM. Melphalan (an alkylating agent) and bortezomib (a proteasome inhibitor) are two of the most common medications used to treat MM, they both work by inhibiting several steps in the DNA repair process that play a role in both the body’s response to therapy and any subsequent resistance [88]. DNA damage leads to the activation of autophagy, which plays a role in DNA damage response (DDR) and assists in cytokine secretion, senescence, cell death, and repairing DNA lesions [89]. DNA damage leads to the activation of ataxia-telangiectasia mutated (ATM) within the Mre11-Rad50-Nbs1 complex (MRN) that binds double-strand breaks. ATM is then able to activate AMPK that targets a TOR-Autophagy Spatial Coupling Compartment (TORC1) inhibitor known as tumor necrosis factor receptor (TNFR)-associated factor interacting protein 2 (TSC2). TORC1 acts as a negative regulator of the ULK complex; therefore, its inhibition allows autophagosome formation. ATM also regulates p53 and stabilizes it, and in turn, p53 regulates other autophagic pathways, including AMPK, PTEN, and damage-associated molecular pattern (DAPK). Furthermore, p53 modulates Sestrin, a protein that regulates TORC1 activity in an AMPK-independent manner through the GATOR2-GATOR1-RAGB/A (GTPase-activating protein activity toward RAGs) signaling pathway as well as via AMPK-dependent manner like how it affects TSC2. ATM activates Che-1, an RNA polymerase II-binding protein that increases the transcription of Deptor and Redd1 as well as inhibiting mTOR. ATM also modulates Beclin1 through the activation of the NF-ϰB essential modulator (NEMO)-dependent transforming growth factor (TAK1)-ATM-NEMO-NFϰB pathway. TAK1 can activate c-Jun N-terminal kinases (JNK), allowing Beclin1 to be released from antiapoptotic proteins. Autophagy also induced DNA damage by decreasing the transcription of MKP-1, a JNK phosphatase; thus, JNK induces autophagy. Moreover, DNA damage triggers an enzyme called PARP1, which in turn decreases cellular ATP and NAD+ levels, activating AMPK and initiating autophagy [89].

Cancer stem cells (CSC) could form cancers and self-renew, which is essential in resistance to anti-cancer agents. Autophagy is associated with the maintenance of CSC pluripotency and increases resistance to cancer drugs [90]. One of the prominent drugs used to treat MM patients is proteasome inhibitors (PIs). PIs cause an increase in the amount of misfolded or unfolded proteins, which increases ER stress. The cells adapt to this stress with the help of multiple pathways, including autophagy, which allows the cells to mitigate stressful environments and survive. Therefore, inducing autophagic cell death may potentially be an additional viable approach to dealing with MM. However, it has been shown that uncontrollable autophagy also promotes drug resistance in myeloma cells, and that blocking autophagy may restore sensitivity to medicines [91,92]. The most potent preclinical studies and clinical trials in MM involving autophagy inhibitor or inducer alone or in combination with bortezomib are summarized in Table 3.

## 4. Role of Apoptosis in Cancer

Apoptosis is a form of programmed cell death or type I programmed cell death, in which the nucleus and cytoplasm of cells shrink, become encased in apoptotic bodies, and engulfed by phagocytic cells [97]. Apoptosis is an essential process for regular development and maintaining tissue homeostasis. Mammalian apoptosis arises through one of two distinct pathways, either the intrinsic or extrinsic pathways. However, both the intrinsic and extrinsic pathways end up with the activation of a particular group of protease enzymes known as caspase proteins which are directly responsible for the final stages of apoptosis, including DNA fragmentation, degradation of proteins, formation of apoptotic bodies, and uptake of the cell via phagocytosis [98]. The intrinsic pathway, also known as the mitochondrial or B cell lymphoma 2 (Bcl-2) regulated pathway, is activated via injury or stress within the cell such as DNA damage, hypoxia, cytokine deprivation, or oncogenic stimulation. The interaction between Bcl-2 family proteins is the hallmark of the intrinsic pathway, which regulates mitochondrial outer membrane permeabilization (MOMP). The members of the Bcl-2 family proteins can be divided into three distinct groups according to function: BH3-only pro-apoptotic proteins (i.e., Bid, Bad, Bik, Bim, Bmf, Puma, Noxa, Hrk/DP5), multidomain pro-apoptotic proteins (i.e., Bak, Bax, and Bok), and anti-apoptotic members (i.e., Bcl-2, Bcl-xL, Mcl-1, Bcl-w, and Bfl-1) [97,99,100].

Intrinsic pathway activation promotes overexpression and activation of BH3-only pro-apoptotic proteins, which could stimulate MOMP directly via activation of multi-domain pro-apoptotic proteins Bax and Bak. Simultaneously, indirect MOMP stimulation also takes a place by antagonizing anti-apoptotic members via heterodimer protein–protein interaction and competing for their binding with Bax and Bak proteins. The homologous BH domains of BH3-only proteins can dock in the elongated hydrophobic grooves of anti-apoptotic members [101]. This is accomplished via the amphipathic α-helix of the BH3 domain that contains four hydrophobic residues (h1-h4) that bind four hydrophobic pockets (P1−P4) within the anti-apoptotic proteins in their BH3 binding groove. For instance, Bim, Puma, and Bid BH3-only pro-apoptotic proteins bind to all the anti-apoptotic members and Bad can bind only to Bcl-2, Bcl-xL, and Bcl-w, whereas Noxa selectively with high affinity binding can inhibit Mcl-1 [102]. Once the apoptotic threshold is reached, pro-apoptotic effector proteins Bax and Bak form oligomers on the mitochondrial outer membrane and form pores. This mediation of MOMP, releases apoptogenic factors cytochrome c and Smac proteins from the mitochondria into the cytosol resulting in downstream caspases and apoptosis activation. Cytochrome c forms the apoptosome in the presence of cofactor Apaf-1, activates caspase-9. Smac inhibits X-linked Inhibitor of Apoptosis Protein (XIAP) [102].

The extrinsic pathway, or death receptor mediated apoptosis, is triggered via extracellular ligands of the tumor necrosis factor superfamily including FasL and TRAIL. This stimulus is transmitted across the membrane, thus recruiting adaptor protein FADD and procaspase-8 and -10 [103]. This results in formation of the death-inducing signaling complex (DISC) which can activate caspase-8 and -10 which can directly induce the downstream executioner caspase such as caspase -3 and -7 to drive full commitment to apoptosis [99]. Moreover, caspases -8 and -10 can cleave and activate Bid, which in turn activates Bak and Bax to induce MOMP, ultimately apoptosis [104]. Thus, the cleavage of Bid is the crosslink between the intrinsic and extrinsic pathways.

Cancer is characterized by the uncontrolled growth and division of cells. Importantly, apoptosis is one of the significant pathways that cancer cells attempt to avoid. One mechanism for cancer pathogenesis is to increase anti-apoptotic proteins while decreasing pro-apoptotic proteins to defect cancer cells apoptosis and cause uncontrolled proliferation.

Moreover, in tumor infiltrated T lymphocytes in the BMM, cancer cells can upregulate death receptor ligands. Normally, CD8+ cells recognize tumor antigens, allowing them to aid in killing of cancer cells. Activated T cells will express FasL to bind Fas, the death receptor involved in the extrinsic apoptotic pathway to initiate the caspase cascade. However, tumor cells can express FasL to evade both the immune response and apoptotic pathway [105]. It was first seen in human leukemia that cancer stem cells showed low expression of Fas and FasL, decreased sensitivity to Fas-induced apoptosis, and greater resistance to chemotherapy [106].

Over 50% of cancers have loss of function mutations in the p53 gene [107]. p53 forms a tetramer in its phosphorylated form to mediate both pro- and anti-apoptotic proteins, where it upregulates and suppresses these proteins, respectively. p53 regulates the Bcl-2 family, as well as expression of Apaf-1 and caspase-6 [108]. Thus, either loss or mutation of p53 results in decreased cancer cells apoptosis.

Additionally, apoptosis will occur in cells that are responsive to the respective chemotherapy, but the cells that are unresponsive to therapy will be able to proliferate and result in chemotherapy resistance. The cells that are responsive to therapy can release damage-associated molecular patterns (DAMPs) that are pro-tumorigenic by shifting the state of tumor-associated macrophages. This will signal immune-silencing and allow cancer cells to proliferate. Interestingly, the death receptors Fas and TRAILR are confirmed to have anti-apoptotic signaling including enhance cell growth, survival, and progression [109], as well as the ability to activate caspase-activated-DNase, causing mutagenesis and DNA damage [110].

In summary, apoptosis plays a critical role in the progression of cancers. Disruptions in the apoptosis process can allow cancer cells to evade death and continue to grow and divide, contributing to the development and maintenance of cancer. Disrupting apoptosis can also allow cancer cells to resist chemotherapy and other forms of treatment, making it more difficult to eliminate the cancer. Thus, induction of apoptosis can be an effective way to treat cancer.

## 5. Apoptotic Pathways in MM

Anti-apoptotic proteins play a major role in the pathogenesis of MM. MM cells exhibit imbalances in their anti-apoptotic protein’s expression levels, especially Mcl-1, that leads to prevent apoptosis and allow continued cell growth by inhibiting and forming heterodimer interaction with Bax and Bak proteins. Mcl-1 is known to be overexpressed in MM and plays a crucial role in MM initiation, progression, and chemoresistance [3]. Remarkably, 52% of newly diagnosed and 81% of relapsed MM patients have shown an increase in Mcl-1 protein expression, which correlates with disease progression and a poor patient survival rate [19]. Furthermore, the most common change in gene expression of MM is Mcl-1 overexpression. Approximately 40% of MM patients have increased expression of Mcl-1 and IL-6 receptors due to a gain or amplification in 1q21 [111]. Many studies currently are focused on advancing treatment therapies for MM and overcoming resistance challenges through inhibition of Mcl-1. Importantly, it has been established that Bcl-2 and Bcl-xL inhibitors such as venetoclax (ABT-199), ABT-737 and navitoclax (ABT-263), which mimics the BH3 domain of Bad and binds Bcl-2 and Bcl-xL, result in upregulation of Mcl-1 and develop resistance. Thus, inhibiting Mcl-1 represent a promising strategy in MM-sensitive and -resistance cells. Another promising avenue for targeting chemotherapeutic resistance via Mcl-1 is through Noxa. Proteasome inhibitors such as bortezomib increase expression of Noxa which selectively targets Mcl-1 for proteasomal degradation [112].

MM cells receive critical signals from the BMM that help them avoid apoptosis and preserve long-term survival (Figure 3). By secreting a set of signaling signals, BM stromal cells (BMSCs) control the expression of anti-apoptotic Bcl-2 family proteins particularly Mcl-1. Transcription of Mcl-1 is regulated via growth factors (e.g., VEGF, EGF), cytokines (e.g., IL-6, IL-5, IL-3), as well as cytotoxic stimuli. MM cells create an IL-6/VEGF loop to interact with BMM. Secretion of IL-6 from MM cells triggers IL-6 release in the BMSCs, inducing more VEGF secretion from the malignant cells and increasing proliferation [113]. In the BMM, MM plasma cells are activated by several factors such as IL-6, JAK/STAT, rat sarcoma/mitogen activated protein kinase (Ras/MAPK), phosphatidylinositol-3 kinase (PI3-K)/Akt, and TNF family including B cell activating factor (BAFF), and a proliferation inducing ligand (APRIL). Additionally, IGF-1 can play a role in the survival of MM cells, which can activate NF-κB and Akt, as well as increase expression of FLIP and cIAP-2 which inhibit caspase-8 [114]. IGF-1 can also downregulate the expression of Bim, resulting in less antagonism of anti-apoptotic proteins [115]. Upregulation of IGF-1 and IL-6 have also been associated as a phenotype of resistance in bortezomib resistant cells [116,117].

## 6. Apoptosis and Autophagy Crosslink in MM

Autophagy and apoptosis are mechanisms to deal with cell damage when cells are under stress. Autophagy is the first response, but if the damage cannot be eliminated, apoptosis is induced. The interplay of these mechanisms is dependent on intensity thresholds or duration of stress. Autophagy will block the activation of apoptosis and in turn, apoptosis will block the action of autophagy through caspase-mediated cleavage of autophagic proteins. Many autophagic and apoptotic proteins in MM have roles in both pathways, allowing the connection between these two mechanisms (Figure 4).

Beclin1 has multiple roles and implications in various human cancers. In mammalian cells, Beclin1 facilitates autophagosome formation and acts as a tumor suppressor. In 40–75% of breast, ovarian, and prostate cancer, Beclin1 gene maps to a tumor-susceptibility locus on human chromosome 17q21 that is monoallelically deleted which increasing the incidence and pathogenesis of tumor formation [68,118,119,120,121]. This suggests that this protein has roles in tumor suppression as well as prevention in tumorigenesis. Mouse studies show Beclin1 heterozygous mice demonstrate a predisposition to lymphoma and hepatocellular carcinoma [69].

Importantly, Beclin1 is regulated via several cofactors including AMBRA1, Bif-1, and UVRAG, as well as the anti-apoptotic proteins. Anti-apoptotic proteins inhibit and sequester Beclin1. This will promote self-survival and crosslink both the autophagic and apoptotic pathways. Bcl-2 has a dual role depending on location. It can act in an anti-apoptotic function at the mitochondria by inhibiting MOMP. Additionally, it can act in direct inhibition of autophagy at the ER by inhibiting Beclin1 or indirectly by binding to AMBRA1 [122]. AMBRA1 is a positive regulator and crucial for autophagy initiation. However, AMBRA1 expression is a negative regulator of apoptosis. AMBRA1 binds mitochondrial Bcl-2 proteins, and during autophagy induction, this binding is disrupted, separating mito-Bcl-2 from AMBRA1, leading to an increased release of Bcl-2 and anti-apoptotic function [123]. In cancer cells, AMBRA1 induces cancer cells to undergo autophagy and inhibits tumor cell apoptosis, and therefore may have a role in drug resistance. UVRAG, one of the cofactors of Beclin1, induces autophagy by increasing PI3K class III interaction with Beclin1. Thus, Bcl-2 represses the function and UVRAG activates the function of Beclin1. In genetic mutations in tumor cells, mutations in Beclin1 or UVRAG can lead to low level of autophagy, promoting tumor formation [69]. Bif-1 (also known as Endophilin B1) binds the UVRAG-Beclin1 complex and activates autophagy and tumor suppression. Suppression of Bif-1 promotes tumor growth and knockout of Bif-1 in mice increases the development of spontaneous tumors. In humans, monoallelic deletion of Bif-1 is reported in mantle-cell lymphoma and decreased Bif-1 expression is reported in gastric carcinoma [124]. Importantly, there has been reports of resistance to bortezomib in MM. One mechanism of resistance is through higher levels of high mobility group box 1 (HMGB1), which promotes autophagy by disrupting the interaction between Beclin1 and Bcl-2 [125,126].

UVRAG a mammalian homolog of yeast VPS38, activates the PI3K class III. Additionally, by attracting class C Vps complexes (HOPS complexes) and Rab7 of the late endosome, UVRAG enhances autophagosome development. Interestingly, UVRAG has anti-apoptotic activity which inhibits tumor therapy-induced apoptosis through interactions with Bax. UVRAG inhibits MOMP by preventing Bax translocation from the cytosol to mitochondria. UVRAG regulates Bax subcellular location in tumor cells, performing cytoprotective actions in the process [127].

MAPK pathway also play an essential role in the crosslink between autophagy and apoptosis. Specifically, p38 MAPK acts as both a positive and negative regulator of autophagy and apoptosis [128]. Autophagy is triggered by p38 MAPK overexpression, whereas apoptosis follows when p38 MAPK is under-expressed. Additionally, JNK, the stress-activated protein kinase also plays a dual role, regulating apoptosis through phosphorylation of c-Jun and aTF-2 resulting in activation of protein-1 (AP-1) complex and expression of Fas/FasL signaling pathway proteins, as well as regulating autophagy by promoting Bcl-2/Bcl-xL and Bim phosphorylation of Bim, allowing dissociation of Beclin1, and ultimately autophagy induction [128].

Numerous elements of autophagy have been linked to the STAT3 signaling pathways. Additional evidence from recent studies suggests that STAT3’s autophagy is affected by its subcellular location. For instance, several autophagy-related genes, including anti-apoptotic proteins, Beclin1, PI3K class III, CTSB, CTSL, PIK3R1, HIF1A, BNIP3, and miRNAs with targets of autophagy modulators, are transcriptionally regulated by nuclear STAT3. In contrast, autophagy is permanently suppressed by cytoplasmic STAT3, which sequesters EIF2AK2 and interacts with other autophagy-related signaling molecules including FOXO1 and FOXO3. Additionally, mitochondrial translocation of the transcription factor STAT3 inhibits autophagy in response to oxidative stress, which may help to protect mitochondria against degradation via mitophagy [129].

ERK and Akt signaling pathways through parallel processes promoted the mTORC1 signaling pathway. mTORC1 complex acts as an important regulator of growth and metabolism. The activity of the complex is regulated through Akt and ERK signaling pathways. Thus, activation of Akt and ERK pathways in MM is a challenge for mTOR inhibitor therapy. However, pp242 mTOR inhibitor can overcome activation of Akt in MM cells but with induction of ERK activation which is still a problem of resistance [130,131,132]. Of note, mTORC1 was reported in several studies to positively regulate Mcl-1 synthesis through translational control [133], while AMPK reported to have the opposite impact [134], suggested that mTORC1 and AMPK the nutrient and energy sensors of the cell, respectively, play a critical role in regulation the synthesis of the short-lived protein Mcl-1.

ATGs are not only important in macroautophagy, but these proteins are also important for apoptosis mechanism, which controls their activity and forms a critical target for cancer. The Atg4 family of endopeptidases regulates autophagosome biogenesis. Specifically, the Atg4 family member (Atg4D) is cleaved by caspase-3 during apoptosis, suggesting that native Atg4D is enzymatically inactive and activity following caspase cleavage is enhanced by autophagy [135]. ATG12 is required for macroautophagy and has known conjugation targets called ATG5 and ATG3. ATG3 is the E2-like enzyme necessary for ATG8/LC3 lipidation during autophagy. ATG12-ATG3 complex formation disrupting does not affect starvation-induced autophagy. Rather, the lack of complex formation inhibits cell death mediated by mitochondrial pathways [136]. Additionally, ATG5 is essential for autophagy induction, but can induce apoptosis under stress. Overexpression of ATG5 results in increased sensitivity to chemotherapy agents, but inhibition results in resistance to chemotherapy. The overexpressed ATG5 is cleaved by calpains, and this cleaved form can be recruited to the mitochondria to bind Bcl-xL and activate apoptosis [137]. Downregulation of ATG5 results in a reduction of autophagy, causing cell proliferation and progression of cancer, as seen in primary melanoma [138].

Controllable autophagy may help eliminate tumor cells and clearly apoptosis prevents the survival of the tumor cells and promotes cell death. However, it been demonstrated that autophagy can become dysregulated and become uncontrollable during cancer, allowing tumor cells to survive and promote chemoresistance. Interestingly, due to the dual function of autophagy, cancer cells can escape programmed cell death and emerge chemo resistant. Additionally, as cancer progresses, the proliferating tumor cells trigger hypoxia, and thus hypoxia-inducible factor (HIF-1) production. HIF-1 act as a transcription factor and increase the transcription of autophagic proteins, leading to increased autophagy and proliferation [74].

Autophagy plays a significant role in plasma cell ontogeny and in the pathophysiology of MM [139]. MM malignant cells are reliant on autophagy for homeostatic mechanisms, high levels of monoclonal immunoglobulin (Ig) are produced which results in a significant amount of misfolded and unfolded proteins. Thus, malignant PCs must rely on the autophagic pathway for and in the pathophysiology of MM by protein degradation. As MM progresses, increased autophagy may regulate apoptosis, therefore lead to the proliferation and growth of PCs [139]. So, the use of autophagic inhibitor may efficiently induce death in MM, synergize with proteasome inhibitors, as well as trigger apoptosis activity, due to the antagonist relationship between apoptosis and autophagy. However, while apoptosis and autophagy are generally thought to antagonize one another, autophagy could also induce apoptosis due to activation of caspase-8 and depletion of apoptosis inhibitors [139].

Excessive and uncontrolled autophagy can reduce cell viability, potentially due to extreme reduction of organelles. Lamy et al. discovered the importance of caspase-10 in promoting the survival of myeloma cells by crosslink between autophagy and apoptosis. IRF4 is a transcription factor common in MM that induces caspase-10 and cFLIP. The constitutive degradation of Bcl-2 associated transcription factor (BCLAF-1) due to caspase-10 associating with cFLIP. This disrupts the interaction of BCLAF-1 with Bcl-2, allowing Bcl-2 and Beclin1 interaction. Inhibition of caspase-10 decreases the degradation of BCLAF-1, increasing the association of Bcl-2 and BCLAF-1 and decreasing the association between Bcl-2 and Beclin1, resulting in autophagic cell death [140]. Thus, BCLAF-1 is required to protect MM cells from uncontrolled autophagy.

## 7. Co-Targeting of Apoptosis and Autophagy in MM

Due to the complexity of autophagy and apoptosis crosstalk, it is poorly understood how to effectively target these mechanisms. Many studies are focused on initially eliminating the pro-survival mechanism of autophagy to promote apoptosis or enhance autophagy to induce autophagic cell death.

Importantly, autophagy represents a pro-survival mechanism, the inhibition of this process can promote the apoptotic effects of drugs. Zeng et al. studied the impact of oridonin on autophagy and apoptosis in MM. They discovered that inhibition of autophagy with 3-MA helped sensitize the cells to oridonin and induce apoptosis. They noted a decrease in Sirtuin1 (SIRT1) and an increase in reactive oxygen species (ROS) intracellularly [141]. Attar-Schneider et al. studied the effects of VEGF inhibitors (Bevacizumab) on MM cell lines (ARP-1 and U266). They showed that VEGF inhibition leads to the upregulation of autophagy to promote cell survival. When the autophagic process is blocked with 3MA, apoptosis increases, leading to higher death rates [104]. Trifluoperazine, an antipsychotic was tested on MM cell lines (U266 and RPMI 8226). When treated with trifluoperazine (TFP), cell growth and autophagy were inhibited, and apoptosis was promoted [142]. Furthermore, when TFP was combined with rapamycin, a potent autophagy inducer, there was reduced cell apoptosis when compared to TFP treatment alone [142]. Never in mitosis-related kinase 2 (NEK2) is a serine/threonine kinase that overexpressed in MM and induces survival and drug resistant [143]. Xia et al. demonstrated that NEK2 enhances MM cell autophagy by stabilizing Beclin1, and a combination of autophagy inhibitor CQ and bortezomib significantly prevents NEK2-induced drug resistance in MM cells [143]. Ikeda et al. discovered a hypoxia-inducible glycolytic enzyme hexokinase-2 (HK2) that contributed to autophagy activation through induction of an anti-apoptotic phenotype in MM cells [144]. When the cells were given hypoxic stress, autophagy was activated and HK2 was upregulated. Thus, 3-bromopyruvate (3-BrPA), HK2 inhibitor, could induce apoptosis under hypoxic conditions alone or as a combination with PIs which could increase cell sensitivity to ER stress, leading to apoptosis in MM cells [144].

Interestingly, Wang et al. studied the effects of autophagy regulating drug on proliferation, apoptosis, and autophagy in MM. Unsurprisingly, autophagy showed a dual role towards RPMI8226 MM cell line. RPMI8226 cell line was treated with rapamycin, autophagy activator, and HCQ and 3-MA, autophagy inhibitors. Rapamycin can inhibit the proliferation associated with induction of apoptosis and autophagy, whereas HCQ can inhibit autophagy and proliferation of RPMI 8226 and induce apoptosis. In contrast, 3-MA can inhibit autophagy, but has almost no effects on proliferation and apoptosis [145].

On the other hand, induction of autophagy demonstrated efficacy towards MM. NEV-BEZ235 was reported as a potent autophagy inducer [146]. MM cell lines (U266, RPMI8226, and KM3) were treated with NEV-BEZ235 caused significant autophagy in all 3 cell lines [146]. Moreover, dihydroartemisinin (DHA), a derivative of artemisinin, can induce autophagy and apoptosis in MM, but the link between the autophagy and apoptosis induced by DHA needs more explanation. However, DHA might induce autophagy and apoptosis through Wnt/β-catenin and P38/MAPK signaling pathway [147].

The compound FTY720 is a potent immunosuppressant which has demonstrated activity against hematological malignancies including MM [148]. FTY720 was observed to induce autophagy which helped to promote apoptosis in U266 cells [148]. Few dysregulated microRNAs (miRNAs) have been studied in the context of MM. Yang et al. reported MIR145-3p’s (miroRNA 145-3p) ability to increase bortezomib sensitivity and to induce autophagy in MM cells [149]. The data suggests MIR145-3p as a potential target in new treatments of MM [149]. Additionally, the expression of miR-137 and miR-197 in MM cell lines were examined for their tumor suppression activity [150]. MM cell lines demonstrated significantly lower miR-137/197 expression, and transfection of miR-137/197 resulted in inhibition of Mcl-1 and induction of apoptosis [150].

Many novel therapies are involved in the AMPK/mTOR pathway, which induces apoptosis via autophagic pathways, and leads to decreased cell proliferation in MM cells. Resveratrol induces AMPK and mTOR signaling inhibition resulting in the inhibition of cell viability through induction of apoptosis and autophagy in MM cell lines (U266, RPMI-8226, and NCI-H929) [151]. Furthermore, the multi-tyrosine kinase inhibitor (TKI) sorafenib induces cell death in MM cells lines (U266, LP2, OPM-2, NCI-H929, RPMI-8226, and Karpas 620) in a caspase dependent and independent manner [152]. In vivo studies under sorafenib treatment, MM cells were able to undergo autophagy, and observed a downregulation of Mcl-1, suggesting that the co-targeting of Mcl-1 by sorafenib and of Bcl-2/Bcl-xL by the ABT-737 improves the efficacy of sorafenib in MM [152].

Overexpression of Mcl-1-1 is frequently seen in hematological malignancies including MM and forms an attractive target. The last decade has seen tremendous advances and various Mcl-1 inhibitors have been developed. Mcl-1 inhibitors may help in overcoming drug resistance and improve treatment of MM by inducing apoptosis and autophagy. Several selective Mcl-1 inhibitors showed promise in treating MM including S64315 or MIK666 (NCT02992483), AMG-176(NCT02675452), AMG-397(NCT03465540), AZD5991(NCT03218683), KS18 [153,154]. Non-selective inhibitor also showed a good effect against MM, obatoclax or GX-15-070 was observed to induce a caspase-independent cell death by induction of autophagy in human oral cancer cells via Mcl-1 inhibition [155]. Mcl-1 inhibitors offer strong anti-myeloma efficacy as a monotherapy in MM; however, most development tactics are centered on combinations. Selective Bcl-2 protein inhibitors and proteasome inhibitors (venetoclax and bortezomib, respectively) have been proven to enhance treatment results when used in combination and several clinical trials are underway. One factor to consider is resistance that may develop with venetoclax administration. This is thought to be due to an increase in the amount of Mcl-1 which neutralizes the ability of venetoclax to induce apoptosis [156]. However, venetoclax in combination with bortezomib or S63845 demonstrated a synergistic effect on MM cells [157,158]. In addition, triple combination treatment with added dexamethasone has been beneficial and represents a promising new approach for dealing with MM [158]. Dexamethasone, a hallmark of MM treatment was used in combination with the Mcl-1 inhibitor S63845 on MM cell lines. It was observed that dexamethasone has a synergistic effect with S63845 which induces apoptosis in MM cell lines [159]. When dexamethasone was replaced with a specific P70S6K1 inhibitor, similar results were obtained with less toxicity on stem and progenitor cells [159]. YM155, a small molecule inhibitor of survivin and Mcl-1 suppressed MM cell growth by inducing apoptosis [160]. Furthermore, YM155 demonstrated similar effects on MM cell lines that were sensitive as well as resistant to bortezomib [160].

## 8. Conclusions and Future Direction

Cancer research has extensively focused on the programmed cell death processes, autophagy, and apoptosis. Apoptosis and autophagy are catabolic mechanisms necessary for organismal growth and homeostasis. They may work independently or might have an impact on one another via linked signaling proteins. This review highlighted the molecular processes underlying the interaction between apoptosis and autophagy, which will be a crucial source of molecular targets for therapeutic applications. Autophagy and apoptosis crosslink plays a significant role in MM progression and drug resistance. Several of the representative studies demonstrated autophagy and apoptosis interaction and how they become dysregulated in various stages of MM progression. The present review concludes that apoptosis and autophagy are additional channels that may confirm MM cell death. Furthermore, MM receives several signals from BMM that alters apoptosis and autophagy pathways which form the key factors to escape cell death and resist chemotherapeutic agents. A potential approach to combat MM cell survival may include inhibiting cytoprotective autophagy and re-educating its route to trigger apoptosis. Thus, targeting autophagy and apoptosis combined with chemotherapeutic agents represent a promising strategy to overcome MM cell survival and drug resistance. Understanding the role of autophagy and apoptosis signaling in MM may provide insight into effective therapies. Additionally, future clinical studies should further examine the anti-myeloma efficacy of treatment combinations that co-target autophagy and apoptosis.

## Figures and Tables

**Figure 1 ijms-24-00644-f001:**
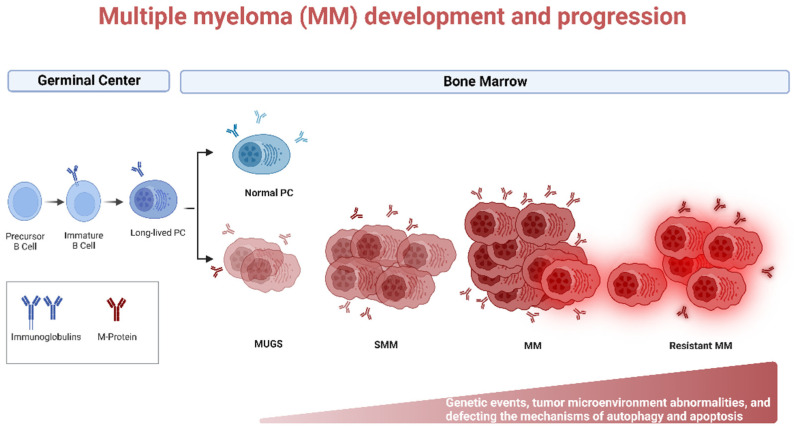
Biology of Multiple myeloma (MM) development and progression. MM is the end stage of a multistep neoplastic transformation of PCs. MGUS and SMM are the precursor stages of MM. Increase in aberrant PCs, Igs, and cytogenetic and tumor microenvironment abnormalities precede the onset of symptoms of MM. Relapsed/refractory (R/R) patients often present with a more aggressive phenotype. Genetic events, tumor microenvironment abnormalities, and defective autophagy and apoptosis may occur during MM progression. Created using Biorender.com (accessed on 1 November 2022).

**Figure 2 ijms-24-00644-f002:**
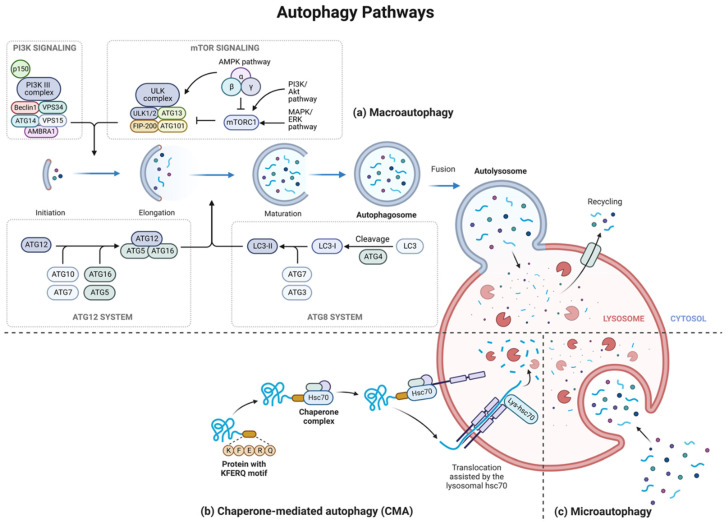
Autophagy pathways in eukaryotic cells is categorized into three main types; (**a**) Macroautophagy machinery is a complex process that involves multiple steps. The initiation step requires the ULK complex, composed of ULK1/2, ATG13, FIP200, and ATG101. Under favorable conditions, such as nutrient-rich environments, mTORC1 inactivates ULK1/2 and ATG13 through phosphorylation. mTORC1 is inhibited under stressful situations and AMP-activated protein kinase (AMPK) complex, which would lead to macroautophagy induction. The nucleation step requires the phosphorylating class III Phosphatidylinositol-3 kinase (PI3K class III) complex, composed of Beclin1, VPS15 and VPS34, ATG14, and AMBRA1. Maturation and elongation process requires the aid of ubiquitin-like proteins and comprises two major steps. First, the ATG12-ATG5 complex is formed from the covalent binding of ATG 12 and ATG5 with the help of ATG7 (E1-like enzyme) and ATG10 (E2-like enzyme). ATG12-ATG5-ATG16L1 complex is formed after ATG16L1 non-covalently binds to the ATG12-ATG5 complex. In the second step, cysteine-like protease- ATG4 cleaves LC3 to form LC3-I. LC3-I is then activated to LC3-II by ATG7 (E1-like enzyme) and ATG3 (E2-like enzyme), followed by its conjugation with lipid phosphatidylethanolamine (PE) with the help of the ATG12-ATG5-ATG16L1 (E3-like enzyme). The phagophore matures to form an autophagosome and finally fuses with the lysosome creating an autolysosome. (**b**) Chaperone-mediated autophagy (CMA)-HSC70 recognizes proteins with KFERQ-like motifs, and with the aid of other chaperone proteins, they deliver the cytoplasmic protein to the lysosome. The protein binds to LAMP2A and gets translocated inside the lysosome with the help of lys-hsc70. (**c**) microautophagy allows cytoplasmic material to enter the lysosome through a cup-like invagination formed in the lysosome’s membrane. Created using Biorender.com (accessed on 12 December 2022).

**Figure 3 ijms-24-00644-f003:**
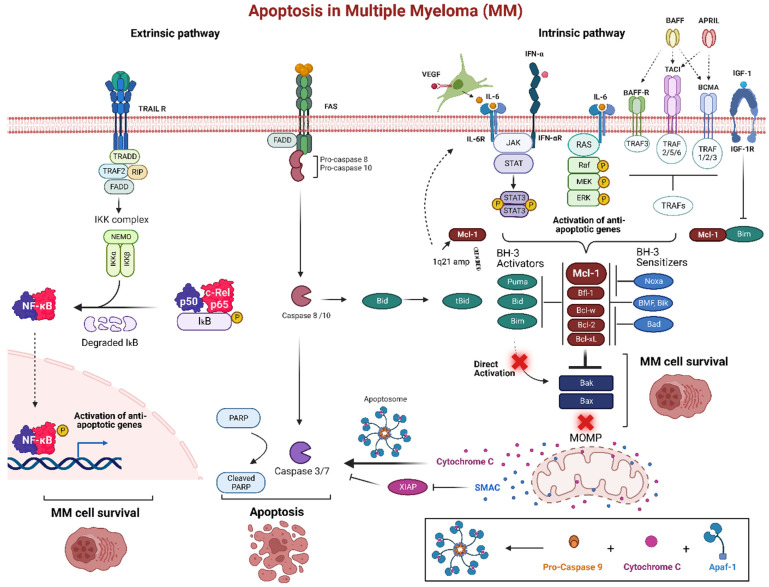
Apoptosis pathway in multiple myeloma (MM). The bone marrow microenvironment (BMM) is an important pathogenic factor for the long-term survival of MM. The intrinsic pathway of programmed cell death is promoted by signaling molecules from the BMM such as IL-6 and IFN-α that trigger various receptors in MM cells. These receptors, including tumor necrosis factor receptor-associated factors (TRAFs) through BAFF-R, BCMA, and TACI, as well as JAK/STAT or Ras/MAPK pathway activate anti-apoptotic proteins, particularly Mcl-1. Many MM patients develop increased expression of Mcl-1 and IL-6 via a gain or amplification of 1q21. This can create an IL-6/VEGF loop, as well as act as a mechanism of resistance through the upregulation of Mcl-1. Overexpression of anti-apoptotic proteins inhibits the BH3-only pro-apoptotic proteins and start competing for Bak and Bax binding site resulting MOMP inhibition. Normally, MOMP stimulation would release cytochrome c, forming the apoptosome after a conjunction with active caspase 9 and Apaf-1, to induce the executioner caspases 3 and 7 for apoptosis. The extrinsic pathway is triggered via the death receptors Fas and TRAIL. These receptors recruit FADD and pro-caspase-8 and 10 and can either directly activate caspase-8 and 10 to induce the downstream executioner caspases or activate Bid to induce MOMP. TRAIL receptor activation also activates anti-apoptotic genes via NF-κB complex. Created using Biorender.com (accessed on 12 December 2022).

**Figure 4 ijms-24-00644-f004:**
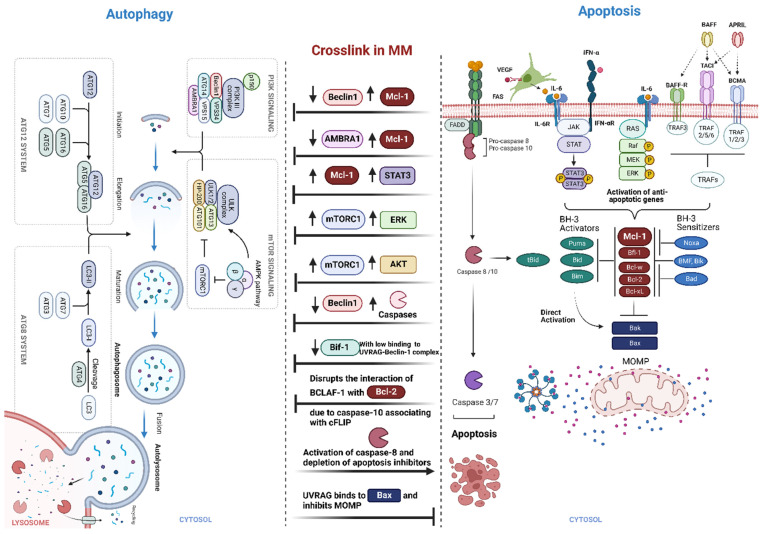
Crosslink between autophagy and apoptosis in MM. Often, the apoptotic pathways can block the action of autophagy. Anti-apoptotic proteins such as Mcl-1, commonly overexpressed in MM, can bind directly to Beclin1, or indirectly to AMBRA1, a regulator of Beclin1. Thus, this binding favors the anti-apoptotic pathway. STAT3 signaling pathways play a critical role in the inhibition of autophagy by regulating several autophagy-related genes including Beclin1 and Mcl-1. Additionally, the PI3K/Akt and MAPK/ERK pathways, members of the apoptotic signaling pathway, can induce the mTORC1 signaling pathway. mTORC1 inhibits the ULK complex, which inhibits the autophagic pathway. Bif-1 normally binds the UVRAG-Beclin1 complex activating autophagy, therefore low expression of Bif-1 promotes cancer cell growth. UVRAG has an additional function that relates the autophagy and apoptosis pathway, as UVRAG can bind Bax and inhibit MOMP, leading to a cytoprotective and anti-apoptotic effect. Suppression of autophagy is also mediated by MM upregulation of caspase-10, which associates with cFLIP, to cleave BCLAF-1. Cleavage of BCLAF-1 does not allow its association with Bcl-2, and thus Beclin1 and Bcl-2 are still bound to each other, inhibiting autophagy. Created using Biorender.com (accessed on 12 December 2022).

**Table 1 ijms-24-00644-t001:** Mechanism of action and underlying drug resistance mechanisms of the common chemotherapeutic agents.

Class	Drugs	Mechanism of Action	Type of Resistance	Mechanism of Resistance
Alkalyting agents Anthracyclines	Melphalan Cyclophosphamide Doxorubicin	Impairment of DNA synthesis and cell replication, immunostimulatory activity by inhibiting interleukin-6 (IL6), interaction with dendritic cells, and immunogenic effects in the tumor microenvironment [4,5]. Topoisomerase II inhibition (Doxorubicin).	Alters autophagy and apoptosis signaling pathways Cancer stem cells and bone marrow microenvironment	Upregulation of anti-apoptotic proteins (Mcl-1, Bcl-2, Bcl-xL) [6,7]. Stem cell-like phenotype with triggering of ALDH1A1 enzymatic activity and upregulation of BTK [8,9]. Increase of cell adhesion molecules (VLA4) [10].
Proteasome inhibitors	Bortezomib Carfilzomib Ixazomib	Inhibition of proteasome activity; inhibition of NF-κβ activity; induction of apoptosis by activation of caspase-8 and -9; upregulation of pro-apoptotic protein Noxa; downregulation of adhesion molecules on myeloma cells [4,11,12,13,14,15,16,17,18].	Alters autophagy and apoptosis signaling pathways Cancer stem cells and bone marrow microenvironment	Upregulation of the proteasomal system; Upregulation of anti-apoptotic proteins (Mcl-1, Bcl-2, Bcl-xL); activation of autophagy pathway; induction of NF-κβ; unfolded protein response (UPR) transcription factor XBP1 suppression; overexpression of heat shock proteins [3,19,20,21,22]. Stem cell-like phenotype with triggering of ALDH1A1 enzymatic activity and upregulation of BTK [8,9]. Secreting a group of extracellular signaling cues including IL-6, growth factors such as vascular endothelial growth factor (VEGF), and Insulin-like growth factor 1 (IGF-1); trigger and modulate multiple keys of the transcriptional pathway including Ras/MAPK, JAK/STAT3, and PI3/Akt; Increase of pro-inflammatory TNF-α; Increase of different cell adhesion molecules; overexpression of CXCR4; overexpression of MARCKS [22,23,24,25,26,27,28].
Immunomodulatory agents	Thalidomide Lenalidomide Pomalidomide	Induction of apoptosis by activation of cspase-8 and -9; interaction with BMM and downregulation of adhesion molecules; affecting cereblon (CRBN) and downstream targets; regulation of T and natural killer (NK) cells activity; anti-angiogenic activity [29,30].	Cancer stem cells and bone marrow microenvironment	Stem cell-like phenotype with triggering of ALDH1A1 enzymatic activity and upregulation of BTK [8,9]. Downregulation of CRBN expression and deregulation of IRF4 expression; increased IL-6 expression and constitutive STAT3 activation [31].
Histone deacetylase inhibitors	Panobinostat Vorinostat	Increasing chromatin structure opening, end with activation of tumor suppressor genes [32,33,34].	Bone marrow microenvironment and disruption intracellular signaling pathways	Regulation of actin cytoskeleton and protein processing in endoplasmic reticulum (triggering of MEK/ERK, PI3K, and FAK pathways) [35].
Monoclonal antibodies	Daratumumab Elotuzumab Isatuximab	Antibody-dependent cellular cytotoxicity (ADCC); complement-dependent cytotoxicity (CDC); modulation of target antigen enzymatic activity; macrophage-mediated pagocytosis; apoptosis via Fcγ receptor-mediated crosslinking; stimulation of immune cells activity, particularly T and NK cells [36,37,38,39,40,41].	Bone marrow microenvironment	Competition by the soluble extracellular forms of CD38 and SLAM7 [42].
Selective Exportin 1 (XPO1) inhibitor	Selinexor	Induces apoptosis through nuclear retention and functional activation of tumor suppressor proteins (TSPs), inhibits NF-κβ, and the translation of oncoprotein mRNAs [43,44,45].	-	-
Corticosteroids	Dexamethasone Prednisolone Methylprednisolone	Induction of apoptosis [46].	Bone marrow microenvironment	Increased secretion of pro-survival cytokines bone marrow microenvironment [4,47].

**Table 2 ijms-24-00644-t002:** Mechanism of action of the most prominent autophagy inhibitors and inducers.

Drug	Mechanism of Action
**Autophagy inhibitors**	
3-Methyladenine (3-MA) Wortmannin LY294002	Class III PI3K inhibitors
Chloroquine (CQ) Hydroxychloroquine (HCQ) Lys05 (CQ derivative)	Lysosomal alkalizer
Bafilomycin A1 Concanamycin A	Vacuolar H⁺-ATPases inhibitors
Elaiophylin 4-Acetylantroquinonol B	Inhibition of autophagy flux
Thymoquinone	Permeabilization of the lysosome membrane
Pepstatin A E64d Leupeptin	Lysosomal proteolysis (hydrolases and proteases) inhibitor
Thapsigargin	Sarco/Endoplasmic reticulum Ca²⁺ ATPase (SERCA) inhibitor
Paclitaxel	VPS34 kinase inhibitor and blocking autophosome-lysosome fusion
PT21	VPS34 Kinase inhibitor
Rapamycin Everolimus Deforolimus Temsirolimus (CCI-779)	mTORC1 inhibitors
**Autophagy inducers**	
AZD8055 Torin1 PP242	mTORC1 and mTORC2 inhibitors
GDC-0980	Class I PI3K and mTORC1/2 inhibitor
CH5132799 GDC-0941	Class I PI3K inhibitor
Metformin Spermidine	AMPK activators
Ibrutinib	BTK inhibitor
Vorinostat (SAHA)	Histone deacetylase inhibitor
Perifosine	AKT inhibitors
Tat-beclin 1 peptide	Autophagy inducing peptide
Resveratrol	Sirtuin 1 and S6 kinase inhibitor

**Table 3 ijms-24-00644-t003:** Preclinical studies and clinical trials of autophagy inhibitor or inducer alone or in combination with bortezomib in MM.

Drug	Study Design	Clinical Trial Status
**Autophagy inhibitors**		
Chloroquine (CQ)	In combination with bortezomib and cyclophosphamide in R/R MM patients	Phase II (NCT01438177)
Hydroxychloroquine (HCQ)	In combination with bortezomib and in R/R MM patients	Phase I/II (NCT00568880)
3-Methyladenine (3-MA)	Human MM cell lines (U266, MM.1S, RPMI8226, and ARH 77)	Preclinical [93]
Bafilomycin A1	In combination with bortezomib in U266 MM cell line	Preclinical [94]
Elaiophylin	Human MM cell lines (U266, RPMI8226, KMS11, and H929)	Preclinical [95]
**Autophagy inducers**		
Metformin	Human MM cell lines (RPMI8226 and U266) and in vivo NOD-SCID murine xenograft MM model	Preclinical [96]

## Data Availability

Not applicable.

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
