# Peer review of "Autophagy and Apoptosis: Current Challenges of Treatment and Drug Resistance in Multiple Myeloma"

_ijms, 2022, doi:10.3390/ijms24010644_

Round 1

Reviewer 1 Report

Al-Odatv et al. give a good overview of the role of autophagy and apoptosis in drug resistance in the context of multiple myeloma. Dysregulation of autophagy and apoptosis have critical roles in the growth and survival of multiple myeloma cells. To date, several studies are underway to better understand the mechanisms of drug resistance acquisition in multiple myeloma. In fact, targeting autophagy and apoptosis combined with chemotherapeutic agents could be a promising strategy to overcome myeloma cell survival and drug resistance, and develop targeted therapies.

This review is well written, and the tables and figures are clearly structured. However, the authors should briefly explain, as they do with autophagy, the role of apoptosis in cancer by adding a short paragraph before the section on apoptotic pathways in MM. In addition, all tables presented in the review must be formatted in one and the same style.

Author Response

Response to Reviewer #1:

Al-Odat et al. give a good overview of the role of autophagy and apoptosis in drug resistance in the context of multiple myeloma. Dysregulation of autophagy and apoptosis have critical roles in the growth and survival of multiple myeloma cells. To date, several studies are underway to better understand the mechanisms of drug resistance acquisition in multiple myeloma. In fact, targeting autophagy and apoptosis combined with chemotherapeutic agents could be a promising strategy to overcome myeloma cell survival and drug resistance, and develop targeted therapies.

This review is well written, and the tables and figures are clearly structured. However, the authors should briefly explain, as they do with autophagy, the role of apoptosis in cancer by adding a short paragraph before the section on apoptotic pathways in MM. In addition, all tables presented in the review must be formatted in one and the same style.

Response: Authors are thankful for the encouraging comments. As suggested, we have now modified the text and kept paragraph about apoptosis before “Apoptotic pathways in MM”. Regarding tables, we formatted in the same style.

Reviewer 2 Report

Omar S. Al-Odatv et al focused their review article on authophagy and apoptosis in Multiple Myeloma. The topic is of interest in the field and, overall the manuscript is well written, figures are informative and references are appropriate. Please find below some comments.

Main comments:

In the last 5 years, some similar articles focusing on the same topic have been published.

2022. Autophagy and the Bone Marrow Microenvironment: A Review of Protective Factors in the Development and Maintenance of Multiple Myeloma (PMID: 35663979)

2020. Autophagy and Myeloma (PMID: 32671780)

2018. Autophagy: A New Mechanism of Prosurvival and Drug Resistance in Multiple Myeloma (PMID: 30196237)

2017. Targeting autophagy in multiple myeloma (PMID: 28599191)

In order to increase the originality of this review I would suggest to add sections dedicated to other types of cell death that occur in Multiple Myeloma and that are related to drug response and resistance, such as Immunogenic Cell Death or Pyroptosis by providing a more comprehensive review. 

Minor comments:

1.            Legend of figure 1: the sentence from “Depending  on the stage …to… referred to as MM” has been entirely copied from the introduction. The authors should revise this section.

2.            “Co-targeting of Apoptosis and Autophagy in MM” paragraph: line 583, the author should remove “One group” and specify the first name author et al. Same in line 586

3. The resolution of each image should be increased

Author Response

Omar S. Al-Odat et al focused their review article on autophagy and apoptosis in Multiple Myeloma. The topic is of interest in the field and, overall, the manuscript is well written, figures are informative, and references are appropriate.

Response: Authors appreciate the reviewer’s encouraging comments.  

Main comments:

Comment No. 1: In the last 5 years, some similar articles focusing on the same topic have been published.

  1. Autophagy and the Bone Marrow Microenvironment: A Review of Protective Factors in the Development and Maintenance of Multiple Myeloma (PMID: 35663979)
  2. Autophagy and Myeloma (PMID: 32671780)
  3. Autophagy: A New Mechanism of Prosurvival and Drug Resistance in Multiple Myeloma (PMID: 30196237)
  4. Targeting autophagy in multiple myeloma (PMID: 28599191)

In order to increase the originality of this review I would suggest to add sections dedicated to other types of cell death that occur in Multiple Myeloma and that are related to drug response and resistance, such as Immunogenic Cell Death or Pyroptosis by providing a more comprehensive review. 

Response:  Authors thank the reviewer for this suggestion. The mentioned reviews above or other reviews in literature either focused on autophagy or apoptosis in drug resistance. Our attempt was to cover both cell death mechanism and its cross link in drug resistance. We propose novel strategies to target both pathways to improve the efficacy of the drugs in the clinic as well as in clinical trials. We did not discuss about necroptosis, pyroptosis, and ferroptosis because still this field is evolving especially in multiple myeloma. 

Minor comments:

Comment 1: Legend of figure 1: the sentence from “Depending on the stage …to… referred to as MM” has been entirely copied from the introduction. The authors should revise this section.

Response: The sentence is revised now.  

Comment 2: Co-targeting of Apoptosis and Autophagy in MM” paragraph: line 583, the author should remove “One group” and specify the first name author et al. Same in line 586

Response: Line 583, 586 and 704 modified as suggested.

Comment 3: The resolution of each image should be increased.

Response: We thank the reviewer for this comment. The resolution of the figures has been increased.